# Revision of the Assassin Bug Genus *Sigicoris* stat. nov. Based on Morphological Study and Molecular Phylogeny (Heteroptera: Reduviidae: Peiratinae) [note 1]

**DOI:** 10.3390/insects13100951

**Published:** 2022-10-19

**Authors:** Yingqi Liu, Hu Li, Wanzhi Cai

**Affiliations:** Department of Entomology and MOA Key Lab of Pest Monitoring and Green Management, College of Plant Protection, China Agricultural University, Beijing 100193, China

**Keywords:** Australian region, lectotype, New Guinea, new species, new synonym, taxonomy

## Abstract

**Simple Summary:**

The members of the Reduviidae family are generally known as “assassin bugs” due to their predatory habits. Peiratinae is one of the largest subfamilies of Reduviidae with a cosmopolitan distribution. Based on the morphological examination on type specimens and other peiratine material from New Guinea, as well as the result of a molecular phylogenetic analysis using *COI*, 16S and 18S genes, the subgenus *Ectomocoris* (*Sigicoris*) Miller, 1958 is elevated to genus level and revised. Four species are recognized and keyed, including three new combinations: *S. brumalis* **comb. nov.**, *S. gracilis* **comb. nov.**, *S. sexguttatus* **comb. nov.** and one new species, *S. dominiqueae* **sp. nov.** The lectotype is designated for *Brachysandalus sexguttatus*. *Pirates concinnus* **syn. nov.** is treated as the junior synonym of *S. sexguttatus* **comb. nov.** The systematic relationships, diagnosis, distribution and habitat of this genus are also discussed.

**Abstract:**

Peiratinae is a cosmopolitan subfamily within Reduviidae with more than 300 known species in 34 genera. There are also some taxa endemic to islands, but their taxonomic status and biology require further study. After examining type specimens of the peiratine species distributed in New Guinea, we found that some of them share many morphological characters, though they were previously assigned in different genera. The phylogenetic analysis based on cytochrome oxidase I, 16S ribosomal RNA and 18S ribosomal RNA genes involving 38 species in 25 genera also supports the result of the morphological study that these species should be involved in a separate genus. In the present study, the subgenus *Ectomocoris* (*Sigicoris*) Miller, 1958 is elevated to genus level, *Sigicoris* **stat. nov.** Three new combinations, *S. brumalis* **comb. nov.**, *S. gracilis* **comb. nov.**, *S. sexguttatus* **comb. nov.** and one new species, *S. dominiqueae* **sp. nov.** are described or redescribed. The lectotype of *Brachysandalus sexguttatus* is designated, and *Pirates concinnus* **syn. nov.** is treated as the junior synonym of *S. sexguttatus* **comb. nov.** A key is provided to separate the four species of this genus. The systematic relationships, diagnosis, distribution and habitat of *Sigicoris* **stat. nov.** are briefly discussed.

## 1. Introduction

Being one of the largest subfamilies in Reduviidae, Peiratinae compromises more than 300 valid species in 34 genera [1,2,3,4,5,6,7,8,9,10,11]. Most peiratine species are predators living on the ground. They are usually nocturnal, and hide in rock crevices, decomposing tree trunks and other cryptic microhabitats during the daytime and become active at night [6]. Phylogenetic studies on Peiratinae have been relatively limited and mainly focus on interspecific relationships within the genus [12,13,14,15,16]. More recently, the use of molecular data has contributed much to clarifying unresolved taxa and carrying out phylogenetic studies, such as the taxonomic revision on the *Bekilya* group based on morphological characters and *COI* genes [4]; comparative mitogenomics research on species of *Peirates* Serville, 1831 [17]; and the phylogenetic analysis and evolutionary study of *Sirthenea* Spinola, 1837 based on both molecular and morphological data [18].

Peiratinae is distributed worldwide but is most speciose in the tropical region [19]. There are also some island endemic genera, such as *Bekilya* Villiers, 1949; *Hovacoris* Villiers, 1964 and *Pseudolestomerus* Villiers, 1964 in Madagascar. New Guinea is a large tropical island with high biodiversity, but a detailed taxonomic study of Peiratinae in New Guinea is lacking, and our knowledge about the habitat and biology of insular peiratine species is relatively poor. Up to now, there have been only eight peiratine species in five genera recorded in New Guinea: *Brachysandalus sexguttatus* Stål, 1866; *Ceratopirates leopoldi* Schouteden, 1933; *Ectomocoris brumalis* Miller, 1958; *E. gracilis* Miller, 1958; *E. mimomyrmix* Miller, 1951; *E. olthofi* Miller, 1958; *Peirates concinnus* Walker, 1873 and *Sirthenea nigronitens* Miller, 1958 [1,20]. Among them, *C. leopoldi* was redescribed by Coscarón [21], and *S. nigronitens* was well-studied in Chłond’s revision of *Sirthenea* of the Old World [22]. Miller established a new subgenus, *Sigicoris*, for *E. brumalis* and *E. gracilis* and mentioned that *Sigicoris* is different from *Ectomocoris* Mayr, 1865 in the head, the femora and the venation of the hemelytron [23]. However, this subgenus was treated as a synonym of *Ectomocoris* in Maldonado-Capriles’s catalogue [1]. Coscarón revised the genus *Peirates*: she mentioned that “*P. concinnus* cannot be assigned to any existing genus” but did not erect a new genus for it [14].

After our examination of the type specimens of these New Guinea species, we found that some of them share many morphological characters, even though currently they belong to different genera, especially those species of which the status has been controversial. In addition to the morphological examination, the molecular phylogenetic analysis of Peiratinae using nuclear and mitochondrial genes was also performed to better understand the taxonomic status of these species and their relationships with other peiratine taxa. In the present study, we elevated *Sigicoris* to the genus level, with three new combinations: *S. brumalis* (Miller, 1958) **comb. nov.**, *S. gracilis* (Miller, 1958) **comb. nov.** and *S. sexguttatus* (Stål, 1866) **comb. nov.** These three species together with one new species, *S. dominiqueae* **sp. nov.**, are redescribed or described and *P. concinnus* is confirmed to be the junior synonym of *S. sexguttatus*. A key is provided to help separate the four species of this genus. The systematic relationships between *Sigicoris* **stat. nov.** and its related genera and the distribution and habitat of this genus are also briefly discussed.

## 2. Materials and Methods

### 2.1. Sampling and DNA Extraction

This study is based on the specimens deposited in the Natural History Museum (NHM), London, UK; Royal Belgian Institute of Natural Sciences (IRSNB: Institut Royal des Sciences Naturelles de Belgique), Brussels, Belgium; Royal Museum for Central Africa (RMCA: Musée Royal de l’Afrique Centrale), Tervuren, Belgium; French National Museum of Natural History (MNHN: Muséum National d’Histoire Naturelle), Paris, France; Swedish Museum of Natural History (NHRS: Naturhistoriska riksmuseet), Stockholm, Sweden; American Museum of Natural History (AMNH), New York, USA; Natural History Museum of Los Angeles County (LACM), California, USA; and China Agricultural University (CAU), Beijing, China. The label information and original depository of voucher specimens representing 31 species of Peiratinae for DNA extraction are provided in Appendix A.

The genomic DNA of specimens was extracted using the QIAamp DNA Micro Kit as per the manufacturer’s protocol. In order to reduce the damage to museum specimens and improve the total quality of DNA, the whole specimen was soaked in tissue lysis buffer after removing from the pin for over 12 hours. After DNA extraction, the specimen was rinsed with deionized water and then pinned back.

### 2.2. Sequencing and Assembly

An Illumina TruSeq library was prepared for each species to ensure the sequencing quality. All libraries were prepared with an average insert size of 350 bp and sequenced using the Illumina HiSeq 2500 platform with 150 bp paired-end reads. Raw reads were trimmed of adapters using Trimmomatic [24]. Prinseq version 0.20.4 [25] was used to remove short and low-quality reads with poly-Ns (>15 bp Ns), or >75 bp bases with a quality score ≤3. The remaining reads were de novo assembled using IDBA-UD [26], with minimum and maximum k values of 45 and 145 bp, respectively.

The contamination and DNA degradation of some museum specimens made the sequencing results of each species uneven. We finally chose three gene fragments with high species coverage viz. the nuclear ribosomal RNA (rRNA) gene 18S and the mitochondrial genes cytochrome oxidase I (*COI*) and 16S rRNA to avoid too much missing data in the dataset. Using the gene fragments of Peiratinae obtained from GenBank as bait sequences, the above three genes of each newly sequenced species were identified from corresponding assemblies by blast searches.

### 2.3. Phylogenetic Analysis

Combining newly sequenced genes and sequences available from GenBank, the ingroup for the phylogenetic analysis included 38 peiratine species representing 25 genera; two other assassin bugs, *Sphedanolestes impressicollis* (Stål, 1861) (Harpactorinae) and *Triatoma rubrofasciata* (De Geer, 1773) (Triatominae), were selected as outgroups (Appendix A). As our examined specimens of *Ectomocoris gracilis* Miller, 1958 were all type specimens and there were no additional specimens for DNA extraction, only the other three putative species of *Sigicoris* **stat. nov.** (*E. brumalis* Miller, 1958, *Sigicoris dominiqueae* **sp. nov.**, and *Brachysandalus sexguttatus* Stål, 1866) were included in the phylogenetic analysis.

Sequences of *COI* were aligned using codon-based multiple alignments under the mafft algorithm [27] implemented in the translatorx online platform with the L-INS-i strategy and default settings [28]. Sequences from each of the two rRNA genes were aligned separately using the mafft v.7.0 online server with G-INS-i strategy [29]. All alignments were then checked manually in MEGA v.7.0. [30]. The dataset, which consisted of 3284 bp, was concatenated with complete *COI* (1533 bp), complete 16S (1280 bp) and partial 18S (471 bp). The 18S fragments of nine peiratine species and partial *COI* fragment of *Ectomocoris quadriguttatus* (Fabricius, 1781) (for details, see Appendix A) were not available from the assemblies and thus treated as missing data.

Phylogenetic trees were constructed under maximum likelihood (ML) methods using IQ-TREE web server [31]. The dataset was partitioned by gene. The best evolutionary model for each partition was selected under the corrected Akaike Information Criterion (AIC) in IQ-TREE as GTR + F+I + G4 (for *COI* and 16S) and TVMe + I+G4 (for 18S). The ML tree was selected with IQ-TREE by an ultrafast bootstrap approximation approach with 10,000 replicates.

### 2.4. Morphological Study

Methods of male genitalia dissection, image capturing and processing followed those of Liu et al. [8,9,10]. Measurements were obtained using a calibrated micrometer. Body length represented the distance between the apex of the head and the tip of the abdomen in resting condition. The distribution map was built using the online version of SimpleMappr [32]. The distribution data were based on our examination of museum specimens, supplemented by data from Miller [23]. Morphological terminology mainly followed Lent and Wygodzinsky [33] and Liu et al. [8,9,10].

## 3. Results

### 3.1. Phylogenetic Analysis

The phylogenetic tree under ML analysis is shown in Figure 1. The reconstruction recovers the monophyly of Peiratinae and the genera with more than two representative species: *Androclus* Stål, 1863; *Peirates* Serville, 1831; *Fusius* Stål, 1862; *Phalantus* Stål, 1863 and *Lestomerus* Amyot and Serville, 1843. *Ectomocoris* Mayr, 1865 is recovered as polyphyletic with respect to *Ectomocoris* (*Sigicoris*). The three putative *Sigicoris* species, *Ectomocoris brumalis*, *Brachysandalus sexguttatus* and the new species described in the present study are grouped together (highlighted in red in Figure 1) with high support (node support value = 100), and they form a separate branch, which is a sister group to the clade (*Lamotteus* Villiers, 1948 + *Parapirates* Villiers, 1959). To better resolve the relationship between *Sigicoris* and *Ectomocoris*, we include the type species of *Ectomocoris*, *E. quadriguttatus*, in the taxon sampling. However, *Sigicoris* is not closely related to the *Ectomocoris* clade; instead, *Ectomocoris* is recovered as the sister group of *Peirates*. As to the genus *Brachysandalus* Stål, 1866, to which *B. sexguttatus* originally belongs, it forms the sister group of the genus *Catamiarus* Amyot and Serville, 1843 and is far from *Sigicoris* in the topology. Hence, the results of the phylogenetic analysis based on molecular data support the genus status of *Sigicoris* **stat. nov.** and further confirm the new combinations of *S. brumalis* **comb. nov.** and *S. sexguttatus* **comb. nov.**

### 3.2. Taxonomy

Order Hemiptera Linnaeus, 1758

Suborder Heteroptera Latreille, 1810

Infraorder Cimicomorpha Leston, Pendergrast and Southwood, 1954

Family Reduviidae Latreille, 1807

Subfamily Peiratinae Amyot & Serville, 1843



**Genus*****Sigicoris*****Miller, 1958 stat. nov.** (Figure 2, Figure 3, Figure 4, Figure 5, Figure 6, Figure 7, Figure 8, Figure 9, Figure 10, Figure 11, Figure 12, Figure 13, Figure 14, Figure 15, Figure 16 and Figure 17)

*Sigicoris* Miller, 1958: 74. As subgenus of *Ectomocoris*; Maldonado-Capriles, 1990: 350. As junior synonym of *Ectomocoris*.

**Type species:** *Ectomocoris* (*Sigicoris*) *gracilis* Miller, 1958, by original designation.

**Diagnosis:** Members of the genus can be recognized among Peiratinae by the following combination of characteristics: head with anteclypeus raised, postocular part ellipsoidal; width of eye shorter than width of interocular space; 1 + 1 tubercles of neck prominent, surface granulose; anterior lobe of pronotum with an elliptical depression on posterior portion, stripes nearly invisible; disc of scutellum broad and granulose, scutellar process slender, apex slightly directed obliquely backward in lateral view; anterior half of metapleural sulcus slightly curved, posterior half nearly straight; ventral surfaces of fore and mid femora with rows of tiny denticles, and each denticle bearing an erect, long seta apically; fore tibia slightly recurved in apical half, with fossula spongiosa occupying at least half of tibial length ventrally; mid tibia with fossula spongiosa occupying at least 1/3 of ventral surface; hemelytron with Cu on corium reduced, Cu and Pcu on membrane short and nearly straight, inner cell relatively short and somewhat quadrilateral.

**Description:** *Coloration*. Mostly brown to blackish brown, sometimes with yellowish white spots on hemelytron and femora and yellowish spots on connexivum.

*Structure.* Macropterous form. Most of body surface covered with scattered setae of varying lengths; costal margin of corium densely covered with short pubescence; ventral surface of abdomen covered with procumbent pubescence (Figure 2, Figure 3, Figure 6, Figure 7, Figure 10, Figure 11, Figure 13, Figure 14 and Figure 15). Head moderately elongated, drop-shaped in dorsal view, anteocular part distinctly longer than postocular; anteclypeus raised; interocular space with a median short sulcus connecting to frontoclypeal sulcus; postocular part ellipsoidal (Figure 4A, Figure 8A, Figure 12A and Figure 16A). Eye oval in dorsal view, width of eye shorter than width of interocular space (Figure 4A, Figure 8A, Figure 13A and Figure 16A); eye reniform in lateral view, reaching dorsal margin but not reaching ventral margin of head (Figure 4B, Figure 8B, Figure 12B and Figure 16B). Ocelli located on small tubercles, width of interocellar space subequal to slightly longer than width of ocellus (Figure 4A, Figure 8A, Figure 12A and Figure 16A). Antenna long and gracile, with first antennal segment thickest and shortest, feebly curved, and second segment straight and the longest. First and second visible rostral segments thick; third segment tapered; second segment longest (Figure 4C, Figure 8C, Figure 12C and Figure 16C). Neck with 1 + 1 tubercles prominent, surface of tubercle granulose (Figure 4A,C, Figure 8A,C, Figure 12A,C and Figure 16A,C).

Pronotum with collar process developed, subconical; length of anterior lobe of pronotum less than twice the length of posterior lobe, anterior lobe with anterior margin almost straight or slightly concave, lateral margins slightly rounded, sculpture nearly invisible; depression on basal of anterior lobe elliptical with a median longitudinal sulcus; pronotal transverse sulcus above with some longitudinal short wrinkles; lateral pronotal angle rounded or narrowly rounded, posterior margin of pronotum convex (Figure 4A, Figure 8A, Figure 12A and Figure 16A). Scutellum triangular, “Y” shaped ridges and scutellar process slender (Figure 4A, Figure 8A, Figure 12A and Figure 16A), apex of process slightly directed obliquely backward in lateral view (Figure 4B, Figure 8B, Figure 12B and Figure 16B), disc of scutellum broad and finely granulose with a median, longitudinal, shallow sulcus (Figure 4A, Figure 8A, Figure 12A and Figure 16A). Stridulitrum long with total-striate type of sculpture (Figure 4C, Figure 8C, Figure 12C and Figure 16C). Pleura and sterna finely granulose (Figure 4B,C, Figure 8B,C, Figure 12B,C and Figure 16B,C); anterior half of metapleural sulcus slightly curved, posterior half nearly straight (Figure 4B, Figure 8B, Figure 12B and Figure 16B); mesosternum with a median longitudinal ridge (Figure 4C, Figure 8C, Figure 12C and Figure 16C). Fore coxa elongate; fore femur not obviously thickened but thicker than mid and hind femora; fore and mid femora with rows of tiny denticles ventrally, and each denticle bearing an erect long seta apically (Figure 4D,E, Figure 8D,E, Figure 12E and Figure 16D,E); fore tibia slightly recurved in apical half, ventral surface with fossula spongiosa occupying at least half of tibial length (Figure 4D, Figure 8D, Figure 12E and Figure 16D); mid tibia with fossula spongiosa occupying at least 1/3 of tibial length (Figure 4E, Figure 8E, Figure 12F and Figure 16E); hind coxae separated from each other less than width of one coxa (Figure 4C, Figure 8C, Figure 12C and Figure 16C). Hemelytron extending beyond tip of abdomen in macropterous male (Figure 2B, Figure 7B and Figure 14B) and nearly reaching tip of abdomen in macropterous female (Figure 3A, Figure 6A, Figure 11A and Figure 15A); Cu on corium reduced, getting fainter from base to apex, Pcu and Cu on membrane short and nearly straight, inner cell relatively short and somewhat quadrilateral (Figure 4F, Figure 8F, Figure 12D and Figure 16F).

Abdomen mainly oval, posterior margin nearly straight with middlemost feebly concave onwards in male (Figure 4G, Figure 8G, Figure 10B, Figure 12H and Figure 14B), tip of abdomen pointed in female (Figure 3B, Figure 6B, Figure 11B, Figure 12G, Figure 15B and Figure 16G); connexivum slightly dilated laterally (Figure 2A, Figure 3A, Figure 6A, Figure 7A, Figure 10A, Figure 11A, Figure 13A, Figure 14A and Figure 15A); venter of abdomen smooth in female (Figure 3B, Figure 6B, Figure 11B, Figure 12G, Figure 15B and Figure 16G) but slightly carinate longitudinally in middle in male (Figure 2B, Figure 4G, Figure 7B, Figure 8G, Figure 13C and Figure 14B). 

Male genitalia asymmetric (Figure 5 and Figure 9). Pygophore oval in ventral view (Figure 5A and Figure 9A); median pygophore process long and slender (Figure 5A–C and Figure 9A–C). Parameres subtriangular (Figure 5D,E) or sickle-shaped (Figure 9D,E), left paramere slightly longer and slenderer than right paramere. Phallus with basal plate bridge feebly curved and slightly longer than basal plate (Figure 5F and Figure 9F). Dorsal phallothecal sclerite broad and flat (Figure 5F and Figure 9F); lateral phallothecal sclerite moderately sclerotized, lower half of inner margin with two processes (Figure 5I and Figure 9I).

**Distribution:** Australian Region (New Guinea) (Figure 17).




**Systematic relationships of *Sigicoris* stat. nov.**


We make the first attempt of the molecular phylogenetic analysis of the whole subfamily, with the taxon sampling containing over 70% of the peiratine genera (25/35) based on the *COI*, 16S and 18S genes. The tree topology helps us further confirm the monophyly and genus status of *Sigicoris* **stat. nov.**, which indicates the application of molecular data in taxonomic revision. *Sigicoris gracilis* **comb. nov.** is not included in the phylogenetic analysis due to the specimen limitation. *Sigicoris gracilis* **comb. nov.** is the type species of *Sigicoris* and can be included in this genus as it shares the morphological characters as described below and the endemic distribution (New Guinea) of the other *Sigicoris* **stat. nov.** species.

*Sigicoris* was first proposed as a subgenus of *Ectomocoris* with two species, *E*. (*Sigicoris*) *brumalis* and *E*. (*Sigicoris*) *gracilis*, while some members of *Sigicoris* **stat. nov.** were originally assigned in *Brachysandalus* or *Peirates*. *Sigicoris* **stat. nov.**, indeed, shares some characters with these three genera, such as the denticles on the ventral surfaces of the fore and mid femora with *Brachysandalus*, the similar length ratio of the anterior lobe of the pronotum to the posterior lobe with *Peirates* and the prominent fossula spongiosa with *Ectomocoris*. However, the reduced Cu on the corium and the short and nearly straight Pcu and Cu resulting in the relatively short and somewhat quadrilateral inner cell on the membrane make *Sigicoris* **stat. nov.** quite unique even in the whole Peiratinae. Except the characters listed above, *Sigicoris* **stat. nov.** can also be distinguished from *Brachysandalus* by the shape of the fore tibia (slightly recurved in the apical half in *Sigicoris* **stat. nov.** vs. clavate and gradually thickened to the apex in *Brachysandalus*), and from *Peirates* by the less distinct and even nearly invisible sculpture on the pronotum. As for the genus *Ectomocoris* to which *Sigicoris* used to belong, the latter could be separated from the former by the smaller eye, the less thickened fore femur, the less elongated anterior lobe of the pronotum and the shape of the abdomen in males (the posterior margin of the abdomen nearly straight with middlemost feebly concave onwards in *Sigicoris* **stat. nov.** vs. the posterior margin of the abdomen arcuate in *Ectomocoris*). In addition, the three sampled *Sigicoris* species are grouped together in our phylogenetic tree and not nested within the *Ectomocoris* clade that includes the type species *E. quadriguttatus*. The results of the morphological comparison and molecular phylogenetic analysis both demonstrate that *Sigicoris* should be separated from *Ectomocoris* as a valid genus.

So far, the only two species in *Ectomocoris* distributed in New Guinea are *E. mimomyrmix* and *E. olthofi*. We examined the apterous female holotype of *E. mimomyrmix* deposited in NHM (label information: “Holotype” disc // red-margined “Type” disc // “PAPUA: Kokoda. 1200 ft. ix. 1933. L. E. Cheesman. B.M. 1934-321.” // “Ectomocoris mimomyrmix sp. n. N.C.E. Miller det. 1950” // “NHMUK 013586064”). Unfortunately, this holotype is badly damaged: the left antenna, the second to fourth segments of the right antenna and most of the legs (except one mid femur and tibia stuck on the card) are missing and the thorax is broken and glued together. Thus, more specimens, especially male and macropterous ones, are needed to further confirm the status of this species. We also examined one male paratype of *E. olthofi* deposited in NHM (label information: yellow-margined “Paratype” disc // “Neth. American New Guinea Exedit. Bernhard camp, 50 m, 2 ix. 1938, J. Olthof” // “Ectomocoris olthofi sp. n. (paratype) N.C.E.Miller det. 1956” // “NHMUK 013585956”) and confirmed its status due to the typical diagnostic characters of *Ectomocoris* such as the anterior lobe of the pronotum more than twice longer than the posterior lobe, the strongly thickened fore femur and the fossula spongiosa nearly occupying the whole ventral surfaces of the fore and mid tibiae. Therefore, currently there are four peiratine genera distributed in New Guinea: *Ceratopirates*, *Ectomocoris*, *Sigicoris* **stat. nov.** and *Sirthenea*.




**Key to the species of *Sigicoris* stat. nov.**


1. Pcu and Cu on membrane slightly curved (Figure 16F); legs and connexivum bicolored (Figure 13, Figure 14 and Figure 15)...................................................................................*S. sexguttatus* **comb. nov.**

– Pcu and Cu on membrane nearly straight (Figure 4F, Figure 8D and Figure 12D); legs and connexivum unicolored (Figure 2, Figure 3, Figure 6, Figure 7, Figure 10 and Figure 11)..................................................................................2

2. First visible segment of rostrum shorter than third segment (Figure 8C); fore tibia with fossula spongiosa occupying half of tibial length (Figure 8D); mid tibia with fossula spongiosa occupying 1/3 of tibial length (Figure 8E).................................*S. dominiqueae* **sp. nov.**

– First visible segment of rostrum longer than third segment (Figure 4C and Figure 12C); fore tibia with fossula spongiosa occupying 2/3 of tibial length (Figure 4D and Figure 12E); mid tibia with fossula spongiosa occupying half of tibial length (Figure 4E and Figure 12F)....................3

3. Coloration blackish brown (Figure 2 and Figure 3); width of interocellar space slightly longer than width of ocellus (Figure 4A)*........................................................**S. brumalis* **comb. nov.**

– Coloration brown (Figure 10 and Figure 11); width of interocellar space subequal to width of ocellus (Figure 12A)*.................................................................................**S. gracilis* **comb. nov.**



***Sigicoris brumalis*****(Miller, 1958) comb. nov.** (Figure 2, Figure 3, Figure 4 and Figure 5)

*Ectomocoris* (*Sigicoris*) *brumalis* Miller, 1958: 75–76. Central West New Guinea, Lake Paniai.

*Ectomocoris brumalis*: Maldonado-Capriles, 1990: 351. 



**Type material examined:** Indonesia: Paratype, 1 male, yellow-margined “Paratype” disc // “Museum Leiden Nieuw Guinen Exp K. N. A. G. 1939 Araboebivak 17. X. 1939” // “Sigicoris brumalis subgen. n. sp. n. (paratype) N. C. E. Miller det. 1956” // “NHMUK 013585911” (NHM); Paratype, 1 female, yellow-margined “Paratype” disc // “Neth. Ind.-Amer. New Guinea Exp. Lower Mist Camp 1500 m 30 S. 1939 L. J. Toxopeus leg.” // “Sigicoris brumalis subgen. n. sp. n. (paratype) N. C. E. Miller det. 1956” // “NHMUK 013585912” (NHM).

**Other material examined:** Indonesia: 1 female, “DUTCH NEW GUINEA: Cyclops Mts. Sabron. Camp 1: 1200 ft. 15. v. 1936. L. E. Cheesman. B.M. 1936-271” // “NHMUK 013585913” (NHM); Papua New Guinea: 1 male, 1 female, “PAPUA: Kokoda. 1200ft. ix. 1933. L. E. Cheesman. B.M. 1934-321” // “NHMUK 013585915” & “NHMUK 013585916” (NHM); 2 males, 2 females, “Stn. No. 46.” // “NEW GUINEA: Madang Dist., Finisterre Mts. Damanti 3550 ft. 2-11. x. 1964” // “M.E. Bacchus. B.M. 1965-120” // “NHMUK 013585917” to “NHMUK 013585920” (NHM). 

**Distribution:** Indonesia (Papua), Papua New Guinea (Oro, Madang).

**Figure 2 insects-13-00951-f002:**
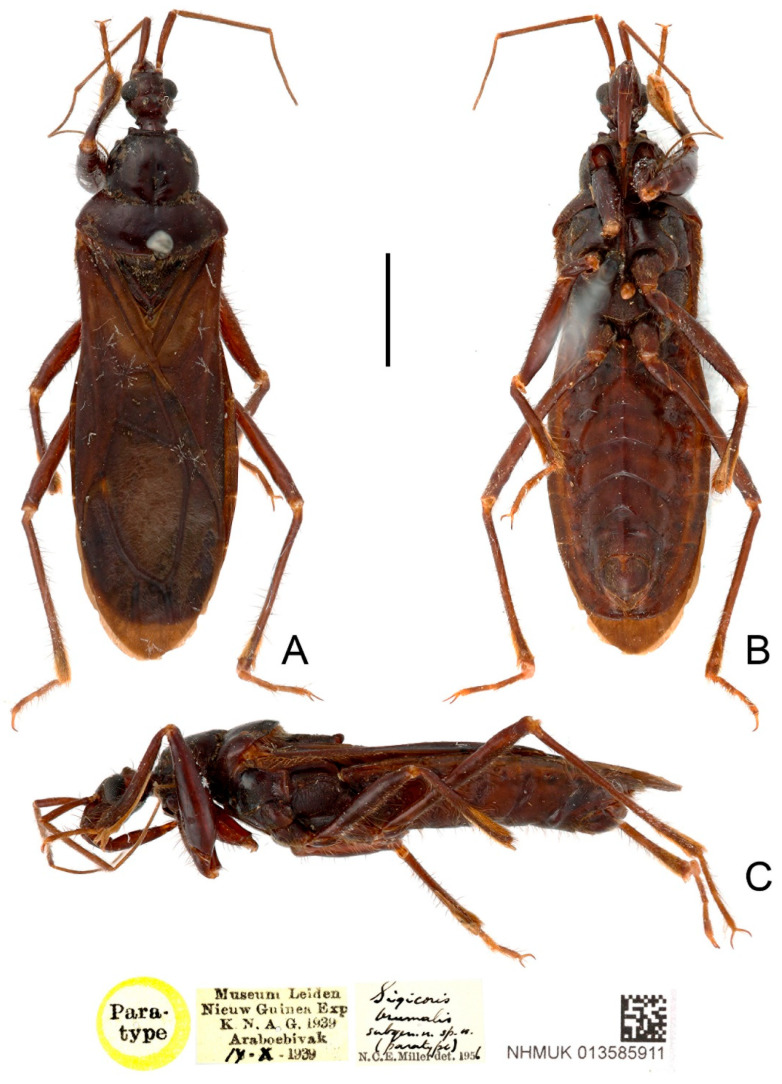
*Sigicoris brumalis* **comb. nov.**, paratype male, habitus. (**A**) Dorsal view. (**B**) Ventral view. (**C**) Lateral view. Scale bar = 3.00 mm.

**Diagnosis:** Body blackish brown in large part; width of interocellar space subequal to width of ocellus; first visible segment of rostrum longer than third segment; pronotum with anterior margin slightly concave, collar process prominent, lateral pronotal angle narrowly rounded; fore tibia with fossula spongiosa occupying 2/3 of tibial length, mid tibia with fossula spongiosa occupying half of tibial length; median pygophore process long, slender and straight, apex sharp, slightly oblique to left side in caudal view, base of inner margin serrated in lateral view; parameres subtriangular with inner margin concave; lower half of inner margin of lateral phallothecal sclerite with two processes, one upward and one downward.

**Figure 3 insects-13-00951-f003:**
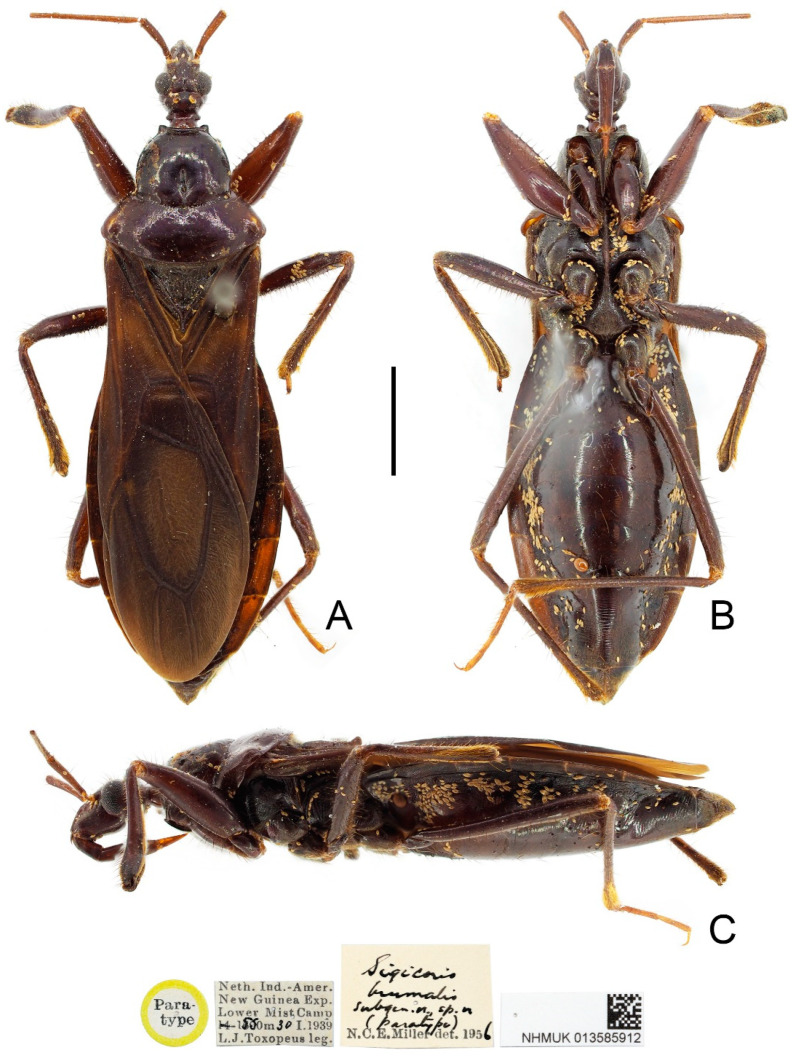
*Sigicoris brumalis* **comb. nov.**, paratype female, habitus. (**A**) Dorsal view. (**B**) Ventral view. (**C**) Lateral view. Scale bar = 3.00 mm.

**Redescription:** Macropterous form (Figure 2 and Figure 3). *Coloration.* Blackish brown. Antennae, third visible segment of rostrum and tarsi brown (Figure 2 and Figure 3); hemelytron dark brown, with most of clavus and area between Pcu and Cu on corium paler, base of costal area of membrane with a diffuse pale brown stripe (Figure 4F).

**Figure 4 insects-13-00951-f004:**
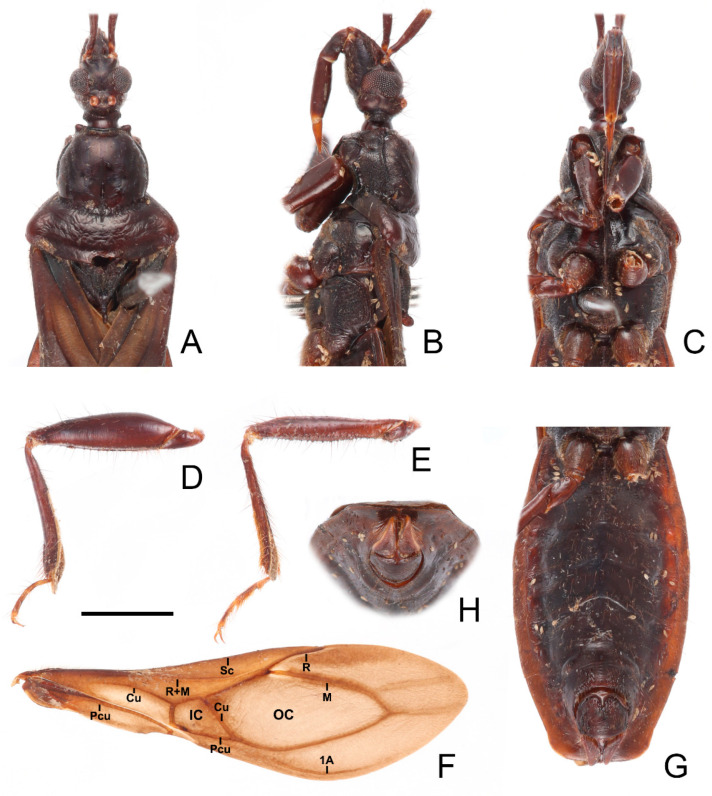
*Sigicoris brumalis* **comb. nov.**, male. (**A**) Anterior part of body with antennae and legs removed, dorsal view; (**B**) ditto, lateral view; (**C**) ditto, ventral view. (**D**) Left fore leg, ventral view. (**E**) Right mid leg, ventral view. (**F**) Right hemelytron, dorsal view. (**G**) Abdomen, ventral view; (**H**) ditto, caudal view. Scale bar = 2.00 mm. Abbreviations: 1A, first analis; Cu, cubitus; M, media; Pcu, postocubitus; R, radius; Sc, subcosta; IC, inner cell on membrane; OC, outer cell on membrane.

*Structure.* Antennae, head, disc of scutellum, pleura, sterna, coxae, ventral surfaces and apex of femora, tibiae, lateral margin of corium, eighth abdominal sternite and genitalic part densely covered with golden, procumbent, short pubescence; first and second antennal segments, lateral margins of head and pronotum, femora and tibiae covered with brown, suberect to erect setae of varying lengths; apex of dorsal surfaces of tibiae and ventral surfaces of tarsi densely covered with yellow to yellowish brown, suberect setae; venter of abdomen sparsely covered with golden, long pubescence (Figure 2 and Figure 3). 

**Figure 5 insects-13-00951-f005:**
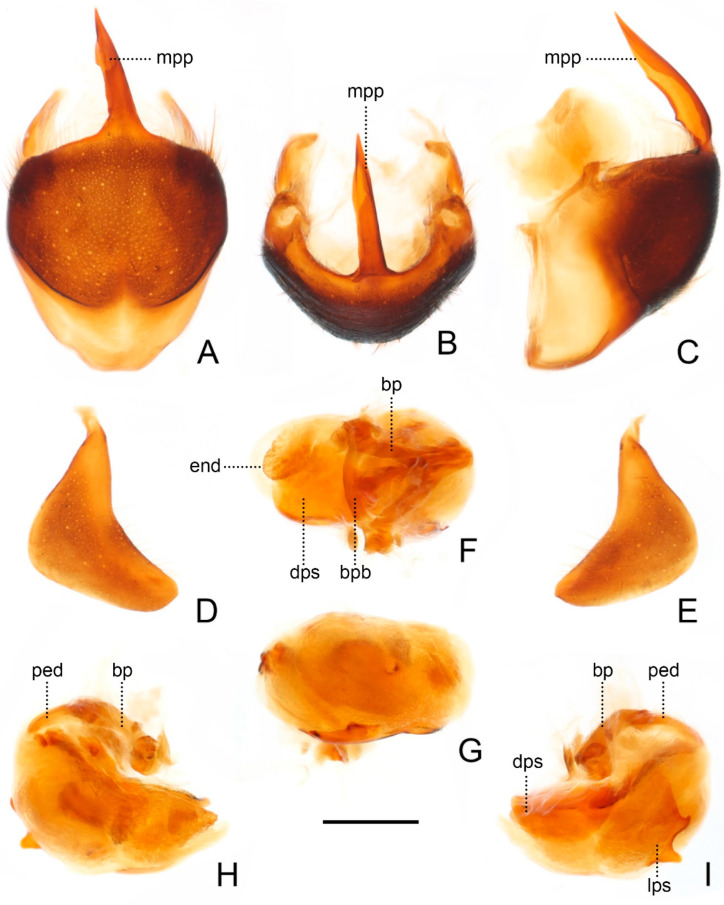
Male genitalia of *Sigicoris brumalis* **comb. nov.** (**A**) Pygophore, ventral view; (**B**) ditto, caudal view; (**C**) ditto, lateral view. (**D**) Left paramere, outer ventrolateral view. (**E**) Right paramere, outer ventrolateral view. (**F**) Phallus, dorsal view; (**G**) ditto, ventral view; (**H**) ditto, left lateral view; (**I**) ditto, right lateral view. Scale bar = 0.50 mm. Abbreviations: bp, basal plate; bpb, basal plate bridge; dps, dorsal phallothecal sclerite; end, endosoma; lps, lateral phallothecal sclerite; mpp, median pygophore process; ped, pedicel.

Head length about 1.29 times as long as width in male and about 1.38 times as long as width in female; median short sulcus on interocular space deep (Figure 4A); width of interocellar space slightly longer than width of ocellus (Figure 4A); first visible segment of rostrum about 1.15 times longer than third segment (Figure 4B). Pronotum with anterior margin slightly concave, collar process prominent (Figure 4A); anterior lobe of pronotum about 1.59 times longer than posterior lobe in male and about 1.40 times longer than posterior lobe in female; lateral pronotal angle narrowly rounded (Figure 4A). Fore tibia with fossula spongiosa occupying 2/3 of tibial length (Figure 4D), mid tibia with fossula spongiosa occupying half of tibial length (Figure 4E).

Male genitalia asymmetric (Figure 5). Pygophore oval in ventral view (Figure 5A); median pygophore process long, slender and straight, apex sharp (Figure 5A–C), slightly oblique to left side in caudal view (Figure 5B), base of inner margin serrated in lateral view (Figure 5C). Parameres subtriangular with inner margin concave, left paramere (Figure 5D) slightly longer and slenderer than right paramere (Figure 5E). Phallus with phallobase strongly sclerotized, basal plate bridge feebly curved and slightly longer than basal plate (Figure 5F); pedicel nearly straight and shorter than basal plate (Figure 5H,I). Dorsal phallothecal sclerite distinctly sclerotized, broad and flat (Figure 5F, H and I); lateral phallothecal sclerite moderately sclerotized, lower half of inner margin with two processes, one upward and one downward (Figure 5I). Apical portion of endosoma with a sacciform process, surface of which is covered with rows of tiny spine-like tubercles (Figure 5F,H).

**Measurements** [in mm, male (*n* = 4), female (*n* = 5)]: Body length 12.68–15.60 (male), 17.22–17.27 (female); maximum width of abdomen 3.54–4.20 (male), 4.56–5.18 (female); head length 1.71–1.74 (male), 1.91–2.08 (female); length of anteocular part 0.79–0.75 (male), 0.95–1.12 (female); length of postocular part 0.42–0.46 (male), 0.45–0.49 (female); head width 1.31–1.38 (male), 1.44–1.45 (female); eye width in dorsal view 0.38–0.47 (male), 0.45–0.41 (female); width of interocular space 0.49–0.58 (male), 0.51–0.62 (female); width of interocellar space 0.29–0.28 (male), 0.32–0.26 (female); lengths of rostral segments I:II:III = 0.73–0.91:1.23–1.51:0.68–? (male), 0.89–0.98:1.42–1.55:0.75–0.82 (female); lengths of antennal segments I:II:III:IV = 1.25–1.42:2.67–2.81:2.33–2.25:1.93–2.42 (male), 1.31–1.59:2.71–2.88:1.95–?: 1.96–? (female); length of anterior pronotal lobe 1.78–2.03 (male), 1.78–2.25 (female); length of posterior pronotal lobe 1.12–1.27 (male), 1.39–1.49 (female); width of anterior pronotal lobe 1.92–2.31 (male), 2.30–2.50 (female); width of posterior pronotal lobe 3.20–3.81 (male), 3.79–4.32 (female); scutellum length 1.33–1.52 (male), 1.78–1.71 (female); maximum width of scutellum 1.56–1.88 (male), 2.02–2.10 (female); hemelytron length 8.98–11.08 (male), 10.36–11.99 (female).

**Remarks:** The holotype of this species is deposited in Rijksmuseum van Natuurlijke Historie, Leiden; we only examined the paratypes deposited in the Natural History Museum, London.



***Sigicoris dominiqueae* Liu, Li and Cai sp. nov.** (Figure 6, Figure 7, Figure 8 and Figure 9).



**Type material:** Indonesia: Holotype female, red-margined “Holotype” disc // “In logs” // “PAPUA: Kokoda. 1200 ft. vS. 1933. L. E. Cheesman. B.M. 1933-577.” (NHM); Paratype 1 female, yellow-margined “Paratype” disc // “Coll. S. R. Sc. N. B. Canopy Mission Papua Neu Guinea (Madang prov): Baiteta 04. VS. 1996 Light trap AR7 Leg. Olivier Missa” orange rectangle label (IRSNB); Paratype 1 male, yellow-margined “Paratype” disc // “OKASA, E. H. D. UNDER BARK FOLLEN HOOP PINE. PINE FOREST. d 3. VIIS. 1967. F. R. WYLIE & S. AUNO” // “123” // “C.S.E.COLL. A. 5212” // “NHMUK 013588800” (NHM).

**Distribution:** Papua New Guinea (Madang, Oro, Eastern Highlands). 

**Etymology:** The specific epithet is dedicated to the French entomologist, Dominique Pluot-Sigwalt (Muséum National d’Histoire Naturelle), in honor of her contributions to entomology and great support to our research. 

**Diagnosis:** Body reddish brown to dark brown, pronotum with anterior lobe blackish brown and posterior lobe dark brown; width of interocellar space longer than width of ocellus; third visible segment of rostrum longer than first segment; pronotum with anterior margin nearly straight, collar process prominent, lateral pronotal angle rounded; fore tibia with fossula spongiosa occupying half of tibial length, mid tibia with fossula spongiosa occupying 1/3 of tibial length; median pygophore process long and extremely slender, apex sharp, nearly vertical in caudal view and curved in lateral view; parameres sickle-shaped; phallobase and dorsal phallothecal sclerite less sclerotized; lower half of inner margin of lateral phallothecal sclerite with two upward processes.

**Figure 6 insects-13-00951-f006:**
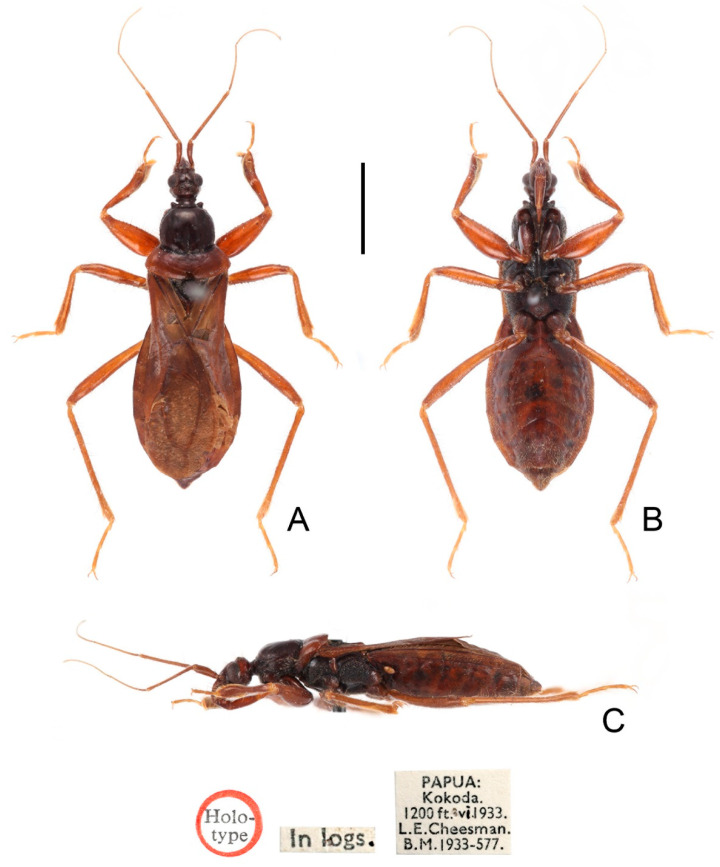
*Sigicoris dominiqueae* **sp. nov.**, holotype female, habitus. (**A**) Dorsal view. (**B**) Ventral view. (**C**) Lateral view. Scale bar = 3.00 mm.

**Description:** Macropterous form (Figure 6 and Figure 7). *Coloration*. Reddish brown to dark brown. Head, anterior lobe of pronotum, scutellum, pleura and sterna blackish brown; third and fourth segments of antenna brown; tarsi yellowish brown (Figure 6 and Figure 7); hemelytron dark brown, with apical half of clavus, area between Pcu and Cu on corium and apical portion of membrane paler, base of costal area of membrane with a diffuse pale brown stripe (Figure 8F).

*Structure*. Antennae, marginal area of head, disc of scutellum, pleura, sterna, coxae, ventral surfaces and apex of femora, tibiae, lateral margin of corium and venter of abdomen densely covered with golden, procumbent, short pubescence; first and second antennal segments, lateral margins of head and pronotum, femora and tibiae covered with brown, suberect to erect setae of varying lengths; apex of dorsal surfaces of tibiae and ventral surfaces of tarsi densely covered with golden, suberect setae; venter of abdomen also sparsely covered with golden, long pubescence (Figure 6 and Figure 7).

**Figure 7 insects-13-00951-f007:**
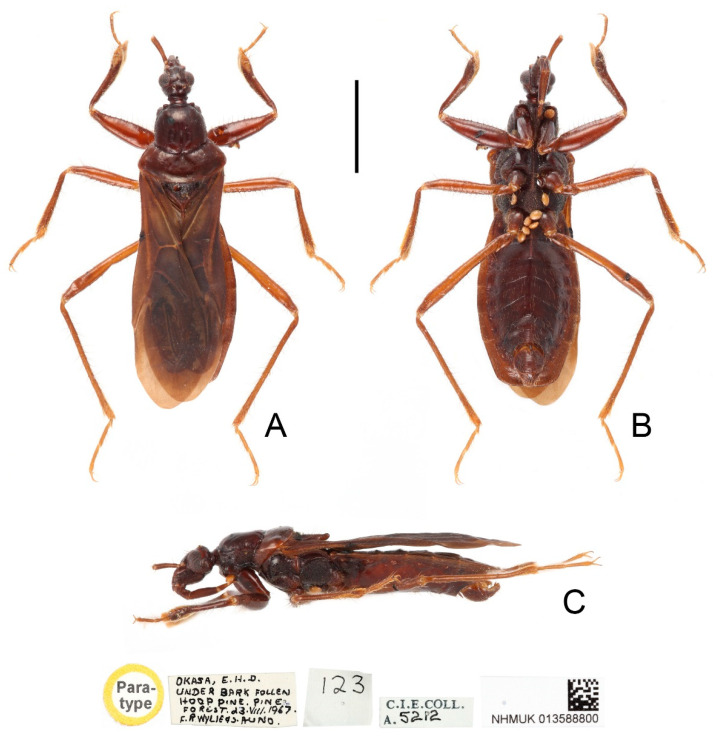
*Sigicoris dominiqueae* **sp. nov.**, paratype male, habitus. (**A**) Dorsal view. (**B**) Ventral view. (**C**) Lateral view. Scale bar = 3.00 mm.

Head length is 1.41 times as long as width in male and about 1.46 times as long as width in female; median short sulcus on interocular space deep (Figure 8A); width of interocellar space longer than width of ocellus (Figure 8A); first visible segment of rostrum shortest, third segment about 1.12 times longer than first segment (Figure 8B). Pronotum with anterior margin nearly straight, collar process prominent; anterior lobe of pronotum about 1.85 times longer than posterior lobe; lateral pronotal angle rounded (Figure 8A). Fore tibia with fossula spongiosa occupying half of tibial length (Figure 8D), mid tibia with fossula spongiosa occupying 1/3 of tibial length (Figure 8E).

Male genitalia asymmetric (Figure 9). Pygophore oval in ventral view (Figure 9A); median pygophore process long and extremely slender, apex sharp (Figure 9A–C), nearly vertical in caudal view (Figure 9B), curved in lateral view (Figure 9C). Parameres sickle-shaped, left paramere (Figure 9D) slightly longer than right paramere (Figure 9E). Phallus with phallobase feebly sclerotized, basal plate bridge feebly curved and slightly longer than basal plate (Figure 9F); pedicel slightly curved and longer than basal plate (Figure 9H,I). Dorsal phallothecal sclerite slightly sclerotized, broad and flat (Figure 9F, H and I); lateral phallothecal sclerite moderately sclerotized, lower half of inner margin with two upward processes (Figure 9I). Apical portion of endosoma with a somewhat petaloid process, surface of which is covered with tiny tubercles (Figure 9F,G).

**Figure 8 insects-13-00951-f008:**
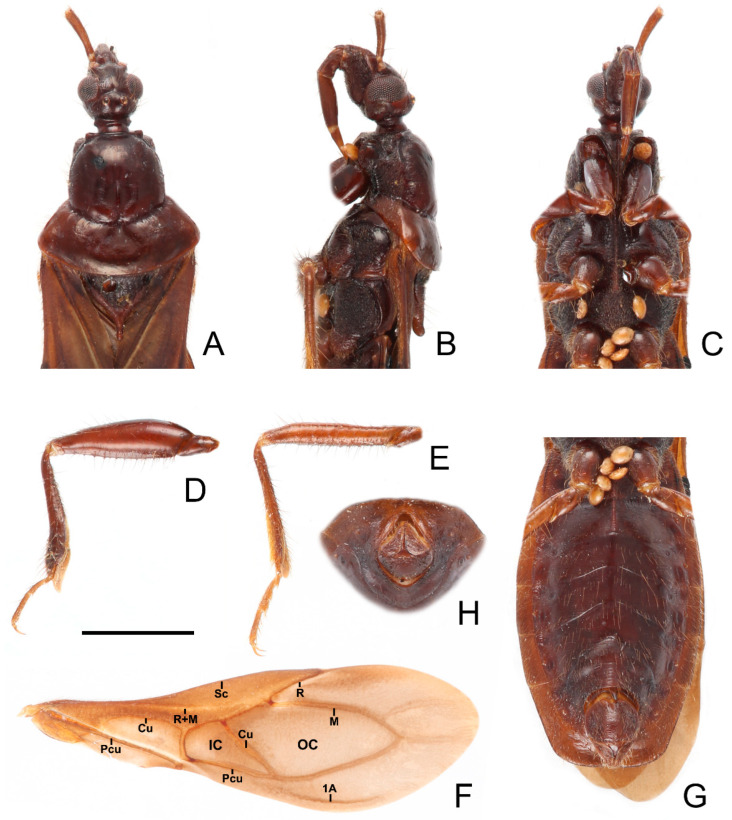
*Sigicoris dominiqueae* **sp. nov.**, paratype male. (**A**) Anterior part of body with antennae and legs removed, dorsal view; (**B**) ditto, lateral view; (**C**) ditto, ventral view. (**D**) Left fore leg, ventral view. (**E**) Right mid leg, ventral view. (**F**) Right hemelytron, dorsal view. (**G**) Abdomen, ventral view; (**H**) ditto, caudal view. Scale bar = 2.00 mm. Abbreviations: 1A, first analis; Cu, cubitus; M, media; Pcu, postocubitus; R, radius; Sc, subcosta; IC, inner cell on membrane; OC, outer cell on membrane.

**Measurements** [in mm, male (*n* = 1), female (*n* = 2)]: Body length 10.56 (male), 10.65–10.70 (female); maximum width of abdomen 3.18 (male), 3.32–3.41 (female); head length 1.62 (male), 1.61–1.60 (female); length of anteocular part 0.78 (male), 0.73–0.78 (female); length of postocular part 0.31 (male), 0.31–0.24 (female); head width 1.15 (male), 1.08–1.12 (female); eye width in dorsal view 0.35 (male), 0.30–0.32 (female); width of interocular space 0.44 (male), 0.48–0.47 (female); width of interocellar space 0.16 (male), 0.12–0.19 (female); lengths of rostral segments I:II:III= 0.57: 0.97: 0.65 (male), 0.58–0.50: 1.01–0.91: 0.68–0.52 (female); lengths of antennal segments I:II:III:IV = 0.94:?:?:? (male), 0.90–0.90:2.01–1.98:1.56–1.35:1.72–1.70 (female); length of anterior pronotal lobe 1.57 (male), 1.55–1.68 (female); length of posterior pronotal lobe 0.90 (male), 0.90–0.80 (female); width of anterior pronotal lobe 1.65 (male), 1.76–1.80 (female); width of posterior pronotal lobe 2.80 (male), 2.70–2.82 (female); scutellum length 1.24 (male), 1.22–1.11 (female); maximum width of scutellum 1.31 (male), 1.25–1.31 (female); hemelytron length 7.50 (male), 6.99–7.29 (female).

**Figure 9 insects-13-00951-f009:**
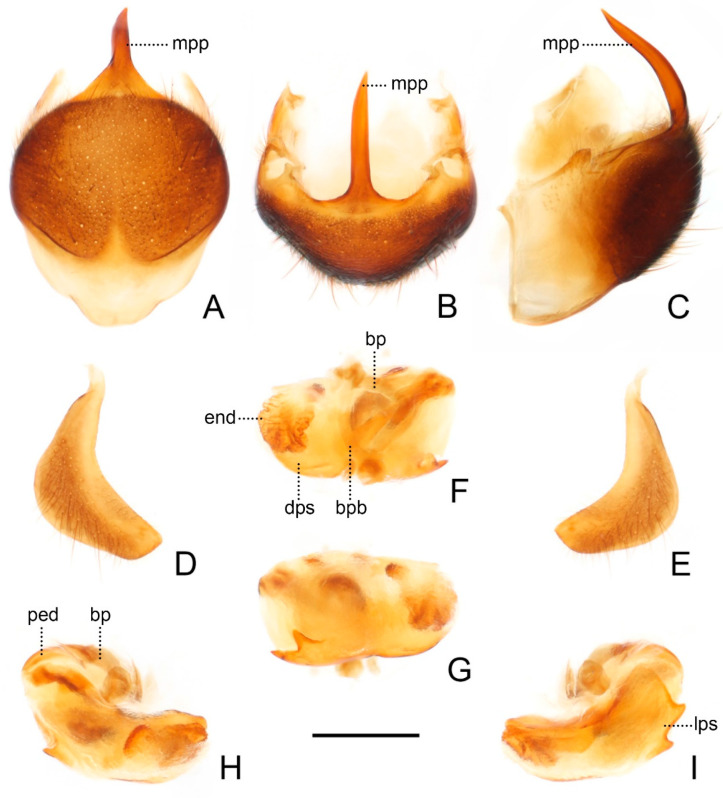
Male genitalia of *Sigicoris dominiqueae* **sp. nov.**, paratype. (**A**) Pygophore, ventral view; (**B**) ditto, caudal view; (**C**) ditto, lateral view. (**D**) Left paramere, outer ventrolateral view. (**E**) Right paramere, outer ventrolateral view. (**F**) Phallus, dorsal view; (**G**) ditto, ventral view; (**H**) ditto, left lateral view; (I) ditto, right lateral view. Scale bar = 0.50 mm. Abbreviations: bp, basal plate; bpb, basal plate bridge; dps, dorsal phallothecal sclerite; end, endosoma; lps, lateral phallothecal sclerite; mpp, median pygophore process; ped, pedicel.

**Remarks:** This new species is similar to *S. brumalis* **comb. nov.** and *S. gracilis* **comb. nov.**, but it can be distinguished from those two species by the following characters: the body size is smaller (less than 10 mm in *S. dominiqueae* **sp. nov.** vs. over 12 mm in *S. brumalis* **comb. nov.** and *S. gracilis* **comb. nov.**), the pronotum is bicolored with the posterior lobe paler (vs. the pronotum unicolored in *S. brumalis* **comb. nov.** and *S. gracilis* **comb. nov.**), the first visible segment of rostrum is shorter than the third (vs. the first visible segment of rostrum longer than the third in *S. brumalis* **comb. nov.** and *S. gracilis* **comb. nov.**) and the fossula spongiosa is less prominent (the fore tibia with the fossula spongiosa occupying half of the tibial length, the mid tibia with the fossula spongiosa occupying 1/3 of the tibial length in *S. dominiqueae* **sp. nov.** vs. the fore tibia with the fossula spongiosa occupying 2/3 of the tibial length, the mid tibia with the fossula spongiosa occupying half of the tibial length in *S. brumalis* **comb. nov.** and *S. gracilis* **comb. nov.**).



***Sigicoris gracilis* (Miller, 1958) comb. nov.** (Figure 10, Figure 11 and Figure 12).

*Ectomocoris* (*Sigicoris*) *gracilis* Miller, 1958: 74–75. Central North New Guinea, Sigi camp.

*Ectomocoris gracilis*: Maldonado-Capriles, 1990: 353.



**Type material examined:** Indonesia: Paratype, 1 female, yellow-margined “Paratype” disc // “Neth. Ind.-Amer. New Guinea Exp. Iebele Camp 1938 2250 mg xi—L. J. Toxopeus leg.” // “Sigicoris gracilis subgen. n., sp. n. (paratype) N. C. E. Miller det. 1956” // “NHMUK 013585907” (NHM); Paratype, 1 male, yellow-margined “Paratype” disc // “Neth. Ind.-American New Guinea Exped. Sigi Camp 1500 m 25 iS. 1939 L. J. Toxopeus” // “Sigicoris gracilis subgen. n., sp. n. (paratype) N. C. E. Miller det. 1956” // “NHMUK 013585908” (NHM); Paratypes, 2 females, yellow-margined “Paratype” disc // “North New Guinea, Hollandia, 140° E. Long. 3°10′ S., 300–600 m, W.S.” // “Sigicoris gracilis subgen. n., sp. n. (paratype) N. C. E. Miller det. 1956” // “B.M. 1938-461” // “NHMUK 013585909” & “NHMUK 013585910” (NHM). 

**Distribution:** Indonesia (Papua).

**Diagnosis:** Body brown with head, pronotum, scutellum, pleura and sterna dark brown; width of interocellar space subequal to width of ocellus; first visible segment of rostrum longer than third segment; pronotum with anterior margin nearly straight, collar process less prominent, lateral pronotal angle narrowly rounded; fore tibia with fossula spongiosa occupying 2/3 of tibial length, mid tibia with fossula spongiosa occupying half of tibial length.

**Redescription:** Macropterous form (Figure 10 and Figure 11). *Coloration*. Brown. Head, pronotum, scutellum, pleura and sterna dark brown; third and fourth segments of antenna and tarsi yellowish brown (Figure 10 and Figure 11); hemelytron brown, with most of clavus, area between Pcu and Cu on corium, most base and marginal area of membrane paler, base of costal area of membrane with a pale brown securiform stripe (Figure 12D).

*Structure*. Antennae, head, disc of scutellum, pleura, sterna, coxae, ventral surfaces and apex of femora, tibiae, lateral margin of corium and venter of abdomen densely covered with golden, procumbent, short pubescence; first and second antennal segments, lateral margins of head and pronotum, femora and tibiae covered with yellowish brown, suberect to erect setae of varying lengths; apex of dorsal surfaces of tibiae and ventral surfaces of tarsi densely covered with golden, suberect setae; venter of abdomen also sparsely covered with golden, long pubescence (Figure 10 and Figure 11).

Head length is 1.24 times as long as width in male and about 1.36 times as long as width in female; median short sulcus on interocular space deep (Figure 12A); width of interocellar space subequal to width of ocellus (Figure 12A); first visible segment of rostrum about 1.37 times longer than third segment (Figure 12C). Pronotum with anterior margin nearly straight, collar process less prominent (Figure 12A); anterior lobe of pronotum 1.58 times longer than posterior lobe in male and about 1.45 times longer than posterior lobe in female; lateral pronotal angle narrowly rounded (Figure 12A). Fore tibia with fossula spongiosa occupying 2/3 of tibial length (Figure 12E), mid tibia with fossula spongiosa occupying half of tibial length (Figure 12F).

**Figure 10 insects-13-00951-f010:**
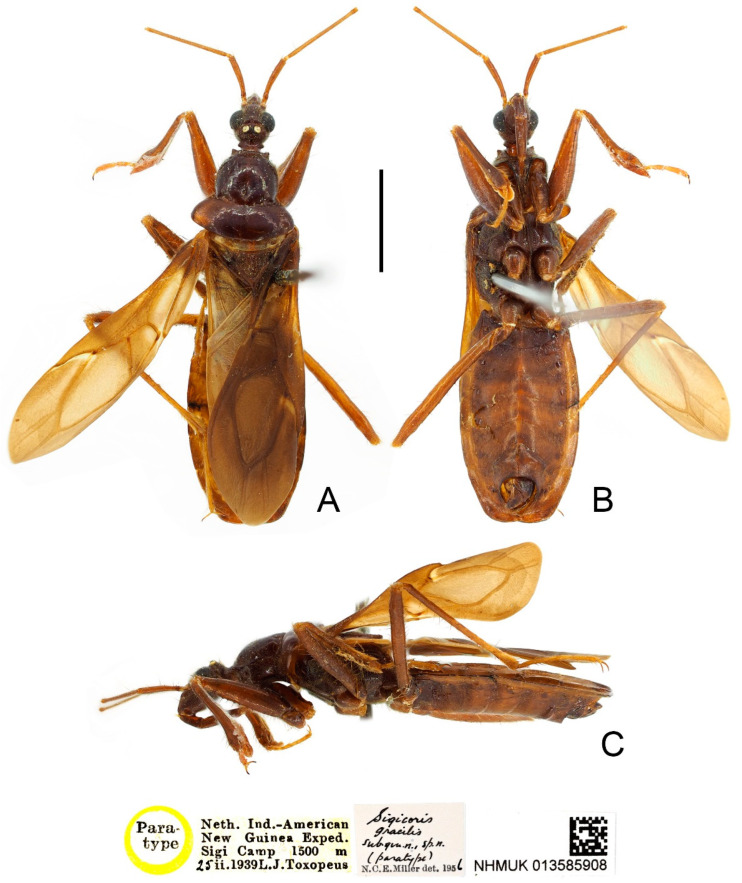
*Sigicoris gracilis* **comb. nov.**, paratype male, habitus. (**A**) Dorsal view. (**B**) Ventral view. (**C**) Lateral view. Scale bar = 3.00 mm.

**Measurements** [in mm, male (*n* = 1), female (*n* = 3)]: Body length 12.19 (male), 14.22–15.36 (female); maximum width of abdomen 3.34 (male), 3.35–4.31 (female); head length 1.59 (male), 1.92–1.93 (female); length of anteocular part 0.70 (male), 0.91–0.92 (female); length of postocular part 0.37 (male), 0.49–0.49 (female); head width 1.28 (male), 1.35–1.50 (female); eye width in dorsal view 0.38 (male), 0.39–0.48 (female); width of interocular space 0.50 (male), 0.59–0.54 (female); width of interocellar space 0.10 (male), 0.18–0.20 (female); lengths of rostral segments I:II:III = 0.76:1.37:0.53 (male), 0.94–1.00:1.52–1.55:0.72–0.73 (female); lengths of antennal segments I:II:III:IV = 1.35:2.50:?:? (male), 1.29–1.38:2.67–2.89:2.13–?:2.51–? (female); length of anterior pronotal lobe 1.58 (male), 1.89–2.05 (female); length of posterior pronotal lobe 1.00 (male), 1.30–1.42 (female); width of anterior pronotal lobe 1.76 (male), 2.11–2.30 (female); width of posterior pronotal lobe 2.91 (male), 3.53–3.81 (female); scutellum length 1.40 (male), 1.29–? (female); maximum width of scutellum 1.52 (male), 1.35–1.85 (female); hemelytron length 8.70 (male), 10.42–10.52 (female).

**Figure 11 insects-13-00951-f011:**
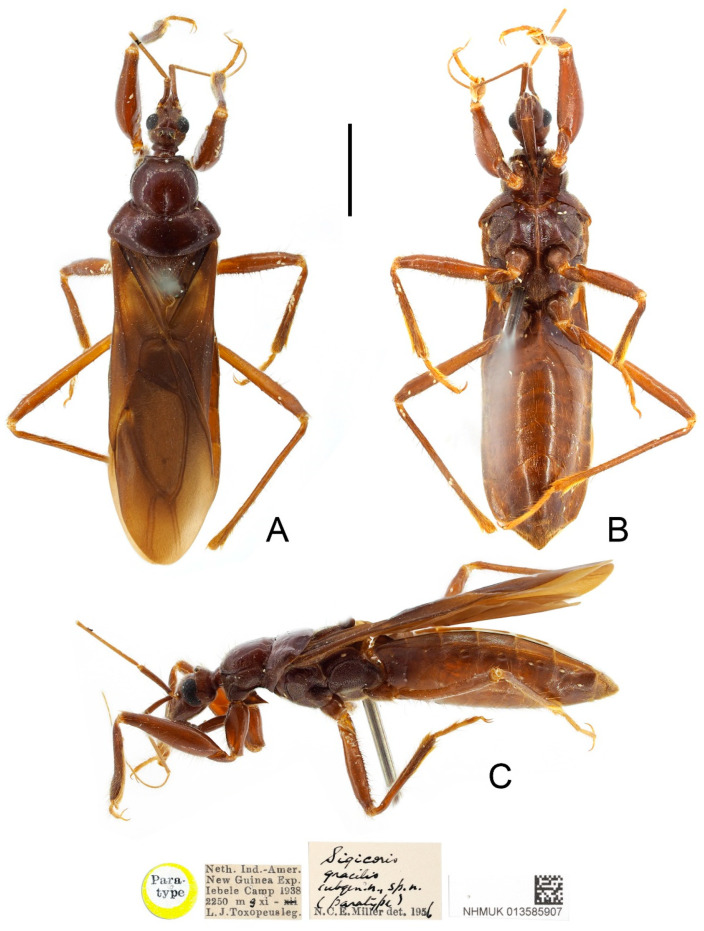
*Sigicoris gracilis* **comb. nov.**, paratype female, habitus. (**A**) Dorsal view. (**B**) Ventral view. (**C**) Lateral view. Scale bar = 3.00 mm.

**Remarks:** The holotype of this species is deposited in Rijksmuseum van Natuurlijke Historie, Leiden. We only examined four paratypes of this species preserved in NHM and there is no additional specimen for dissection and DNA extraction. This species is very allied to *S. brumalis* **comb. nov.**; the characters currently used to separate it from the latter are the smaller body size, the paler coloration, the relatively larger ocellus, the straighter anterior margin of the pronotum with the less prominent collar process and the differences in the male genitalia as Miller illustrated [23]. Here, we still treat them as two valid species, but more evidence such as molecular data and carefully dissected structures of male genitalia are needed to further clarify the relationship between them.

**Figure 12 insects-13-00951-f012:**
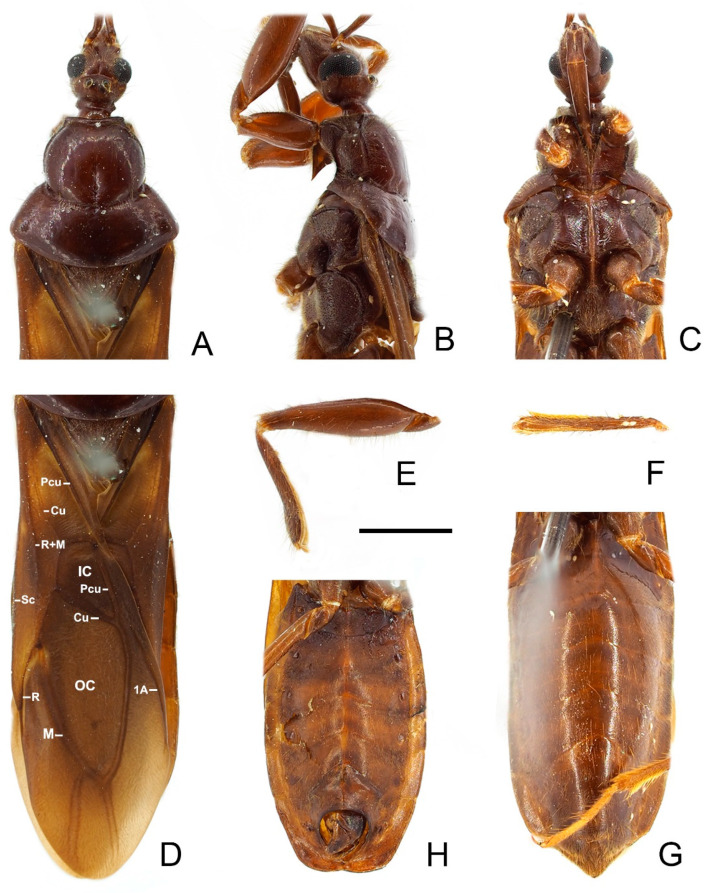
*Sigicoris gracilis* **comb. nov.**, paratypes. (**A**) Anterior part of body with antennae and legs removed, female, dorsal view; (**B**) ditto, lateral view; (**C**) ditto, ventral view. (**D**) Hemelytra, female, dorsal view. (**E**) Left fore leg without tarsus, female, ventral view. (**F**) Right mid tibia, female, ventral view. (**G**) Abdomen, female, ventral view. (**H**) Abdomen, male, ventral view. Scale bar = 2.00 mm. Abbreviations: 1A, first analis; Cu, cubitus; M, media; Pcu, postocubitus; R, radius; Sc, subcosta; IC, inner cell on membrane; OC, outer cell on membrane.

***Sigicoris sexguttatus* (Stål, 1866) comb. nov.** (Figure 13, Figure 14, Figure 15 and Figure 16).

*Brachysandalus sexguttatus* Stål, 1866: 261; Maldonado-Capriles, 1990: 345. Insula Mysol (Indonesia: Misool Island).

*Pirates sexguttatus*: Walker, 1873: 123.

*Pirates concinnus* Walker, 1873: 124. New Guinea. **syn. nov.**

*Pirates* (*Brachysandalus*) *sexguttatus*: Stål, 1874: 60. [34]

*Peirates concinnus*: Maldonado-Capriles, 1990: 364; Coscarón, 1997: 39, 41, excluded from *Peirates* without giving a new status.



**Type material examined:** Indonesia: Lectotype of Brachysandalus sexguttatus (designated by present study), male, “Typus” red rectangle label // “Stevens.” // “Mysol” // “sexguttatus Stål” // “NHRS-GULI 000000135” (NHRS); Holotype of Pirates concinnus, male, red-margined “Holotype” disc // green-margined “Type” disc // “N” disc // “Saunders. 65·13.” // “93. PIRATES CONCINNUS.” // “NHMUK 013587668” (NHM). 

**Other material examined:** Indonesia: 1 ex (without abdomen), “Charles Lewis Mt. New Guinea. 98-203.” // “NHMUK 013587669” (NHM); 1 female, “Indonesia, West Papua ARFAK MTS, 1190 m alt DUEBEI ENV, 21. 1-8.2.2008 cca 20 km S of Warmere Manokwari distr, St Jakl lgt” (CAU).

**Distribution:** Indonesia (West Papua, Papua).

**Figure 13 insects-13-00951-f013:**
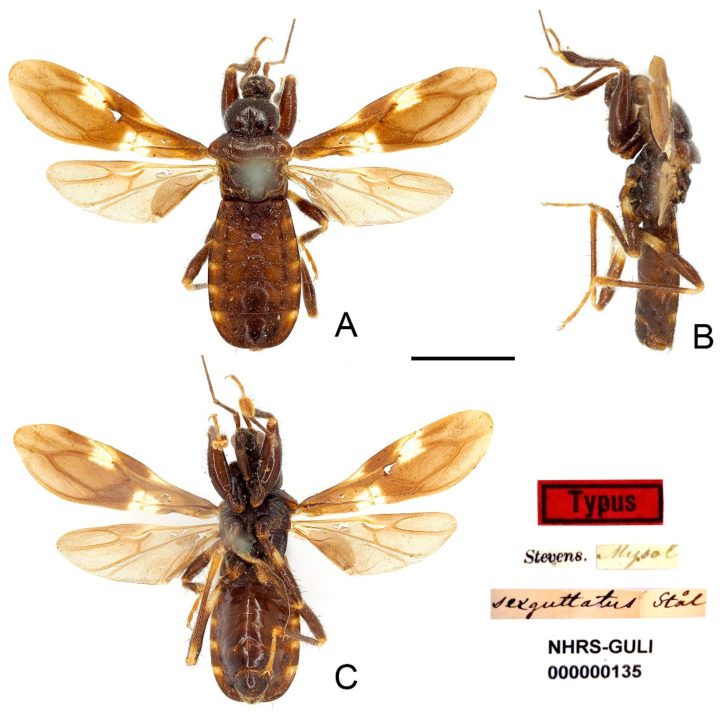
*Sigicoris sexguttatus* **comb. nov.**, lectotype male, habitus. (**A**) Dorsal view. (**B**) Lateral view. (**C**) Ventral view. Scale bar = 3.00 mm.

**Diagnosis:** Body dark brown, base of mid and hind tibiae and basal 1/3 of mid and hind femora yellowish white to pale yellow, hemelytron dark brown with three yellowish white spots, segment of connexivum with basal half yellowish white to yellow and apical half dark brown; width of interocellar space subequal to width of ocellus; first visible segment of rostrum longer than third segment; pronotum with anterior margin nearly straight, collar process prominent, lateral pronotal angle rounded; fore tibia with fossula spongiosa occupying half of tibial length, mid tibia with fossula spongiosa occupying 2/5 of tibial length.

**Redescription:** Macropterous form (Figure 13, Figure 14 and Figure 15). *Coloration*. Dark brown. Head, anterior lobe of pronotum, scutellum, pleura and sterna blackish brown; antennae brown; base of mid and hind tibiae and basal 1/3 of mid and hind femora yellowish white to pale yellow, tarsi yellowish brown (Figure 13, Figure 14 and Figure 15); hemelytron dark brown except area between Pcu and Cu on corium and apical portion of membrane paler, three yellowish white spots present on hemelytron: a trapezoidal spot on conjunctive area of clavus, corium and membrane, a small, narrow, triangular spot on base of area between Sc and M on corium and a somewhat rectangular spot on base of costal area of membrane (Figure 16F); segment of connexivum with basal half yellowish white to yellow and apical half dark brown (Figure 13, Figure 14, Figure 15 and Figure 16G).

**Figure 14 insects-13-00951-f014:**
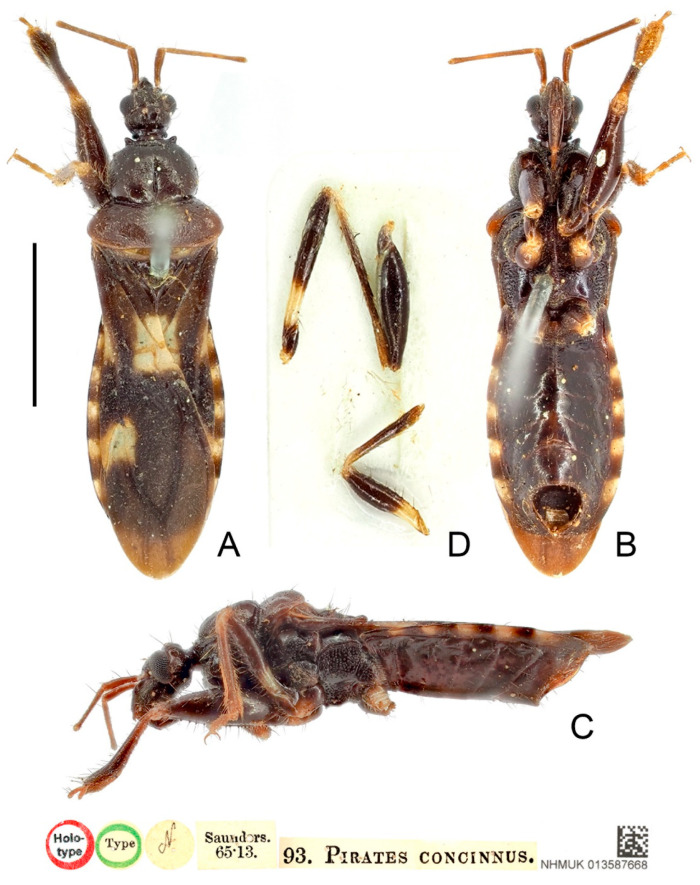
*Sigicoris sexguttatus* **comb. nov.**, holotype of *Pirates concinnus*, male. (**A**) Habitus in dorsal view. (**B**) Habitus in ventral view. (**C**) Habitus in lateral view. (**D**) Legs stuck on the card. Scale bar = 3.00 mm.

*Structure*. Antennae, marginal area of head, disc of scutellum, coxal cavities, coxae, ventral surfaces and apex of femora, tibiae, lateral margin of corium and genitalic part of abdomen densely covered with golden, procumbent, short pubescence; first and second antennal segments, lateral margins of head and pronotum, femora and tibiae covered with brown, suberect to erect setae of varying lengths; apex of dorsal surfaces of tibiae and ventral surfaces of tarsi densely covered with yellow to yellowish brown, suberect setae; venter of abdomen sparsely covered with golden, long pubescence (Figure 13, Figure 14 and Figure 15).

Head length is 1.21 times as long as width in male and 1.28 times as long as width in female; median short sulcus on interocular space deep (Figure 16A); width of interocellar space subequal to width of ocellus (Figure 16A); first visible segment of rostrum about 1.18 times longer than third segment (Figure 16B). Pronotum with anterior margin nearly straight, collar process prominent (Figure 16A); anterior lobe of pronotum 1.62 times longer than posterior lobe in male and 1.39 times longer than posterior lobe in female, lateral pronotal angle rounded (Figure 16A). Fore tibia with fossula spongiosa occupying half of tibial length (Figure 16D), mid tibia with fossula spongiosa occupying 2/5 of tibial length (Figure 16E); hemelytron with Pcu and Cu on membrane slightly curved, not so straight as those in other species of *Sigicoris* **stat. nov.** (Figure 16F). 

**Figure 15 insects-13-00951-f015:**
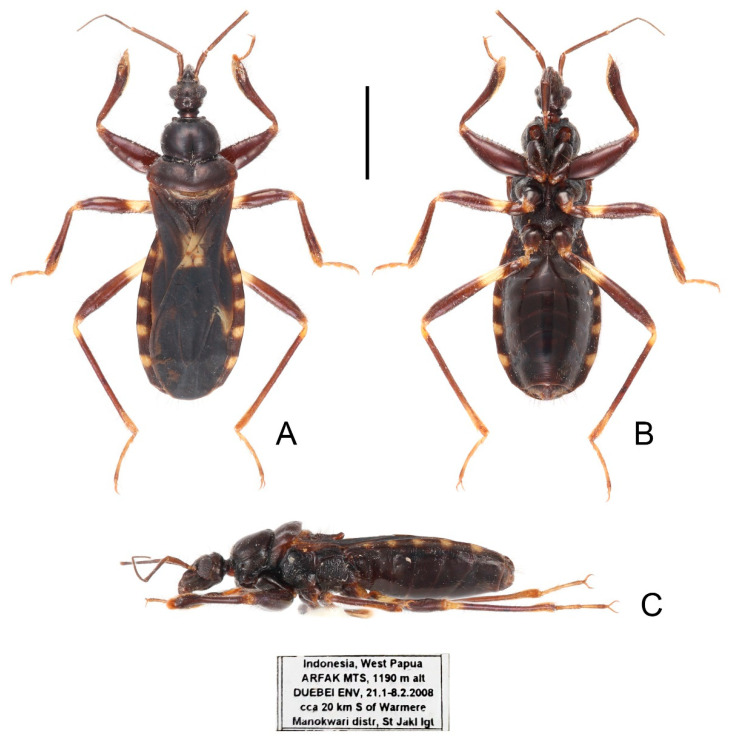
*Sigicoris sexguttatus* **comb. nov.**, female, habitus. (**A**) Dorsal view. (**B**) Ventral view. (**C**) Lateral view. Scale bar = 3.00 mm.

**Measurements** [in mm, male (*n* = 1), female (*n* = 1)]: Body length 8.16 (male), 10.58 (female); maximum width of abdomen 2.44 (male), 3.31 (female); head length 1.22 (male), 1.59 (female); length of anteocular part 0.59 (male), 0.69 (female); length of postocular part 0.28 (male), 0.35 (female); head width 1.00 (male), 1.24 (female); eye width in dorsal view 0.29 (male), 0.35 (female); width of interocular space 0.41 (male), 0.55 (female); width of interocellar space 0.11 (male), 0.18 (female); lengths of rostral segments I:II:III = 0.52:0.91:0.49 (male), 0.79:0.95:0.61 (female); lengths of antennal segments I:II:III:IV = 0.73:1.64:?:? (male), 0.89:1.91:1.35:1.09 (female); length of anterior pronotal lobe 1.38 (male), 1.45 (female); length of posterior pronotal lobe 0.85 (male), 1.04 (female); width of anterior pronotal lobe 1.63 (male), 1.85 (female); width of posterior pronotal lobe 2.40 (male), 2.93 (female); scutellum length 0.88 (male), 0.98 (female); maximum width of scutellum 1.15 (male), 1.35 (female); hemelytron length 6.01 (male), 7.11 (female).

**Figure 16 insects-13-00951-f016:**
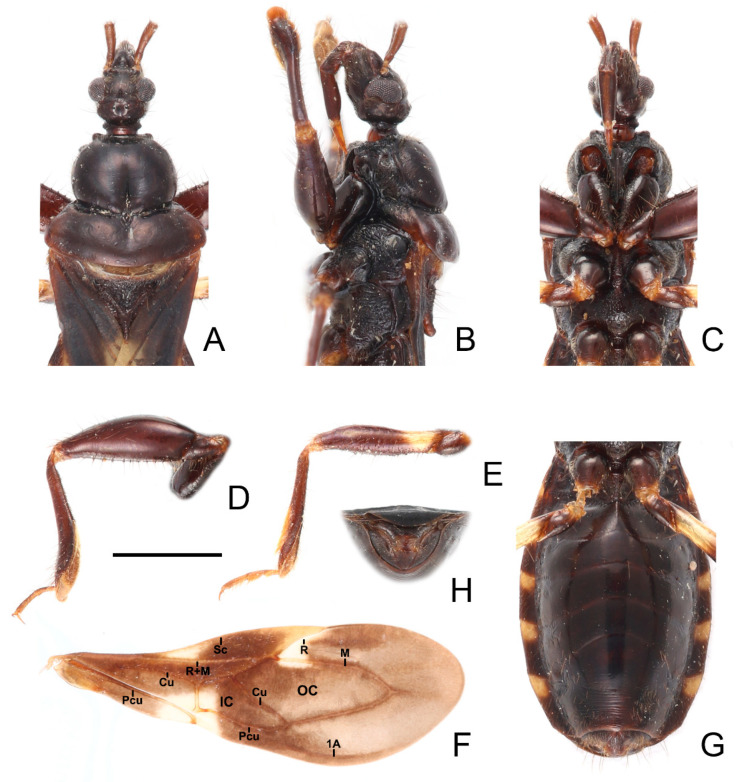
*Sigicoris sexguttatus* **comb. nov.**, female. (**A**) Anterior part of body with antennae and legs removed, dorsal view; (**B**) ditto, lateral view; (**C**) ditto, ventral view. (**D**) Left fore leg, ventral view. (**E**) Right mid leg, ventral view. (**F**) Right hemelytron, dorsal view. (**G**) Abdomen, ventral view; (**H**) ditto, caudal view. Scale bar = 2.00 mm. Abbreviations: 1A, first analis; Cu, cubitus; M, media; Pcu, postocubitus; R, radius; Sc, subcosta; IC, inner cell on membrane; OC, outer cell on membrane.

**Remarks:** The original data of the examined specimens of *Brachysandalus sexguttatus* are “♂, Long. 8, Lat. 2 mill. Patria: Insula Mysol. (Mus. Holm.)”. Stål did not designate the holotype and did not mention the number of specimens he examined [35]. We only found one male syntype of this species in NHRS and its label information matches the original data (Figure 13). Here, we designate this specimen as the lectotype of *B. sexguttatus*.

This species can be easily distinguished from other species of *Sigicoris* **stat. nov.** by the slightly curved Pcu and Cu on the membrane, the yellowish white spots on the hemelytron and the basal 1/3 of mid and hind femora yellowish white. 

## 4. Discussion

### Distribution and Habitat of *Sigicoris* stat. nov.

*Sigicoris* **stat. nov.** is endemic to New Guinea and mainly occurs in the mountainous area, especially along the Central Range which is cross-island from northwest to southeast (Figure 17). *Sigicoris brumalis* **comb. nov.** is the most widespread species within *Sigicoris* **stat. nov.**, which occurs almost transversely across the main island. *S. dominiqueae* **sp. nov.** is only distributed in Papua New Guinea, while *S. gracilis* **comb. nov.** and *S. sexguttatus* **comb. nov.** are only recorded from West Papua Province and Papua Province in Indonesia. Except the main island, the type locality of *S. sexguttatus* **comb. nov.** is Mysol Island (now Misool Island), which is an offshore island in West Papua with the highest point 561 m, and this is also the known westmost distribution of this genus. 

The labels of most of the examined specimens involved altitudinal information. The altitude ranges of most *Sigicoris* **stat. nov.** species are relatively wide: from 300 m to 2250 m. The label information of *S. dominiqueae* **sp. nov.** specimens such as “light trap”, “In logs” and “Under bark fallen hoop pine. Pine forest” could partly reflect the habitat and biology of this species. It can be inferred that species of this genus mainly live in forests; usually hide under barks, decaying logs or other shelters during daytime; become active at night; and can be attracted by the light. The above information expands our knowledge of insular species of Peiratinae to a certain extent, but deeper observation and research are still needed to explain some special biological phenomenon.

**Figure 17 insects-13-00951-f017:**
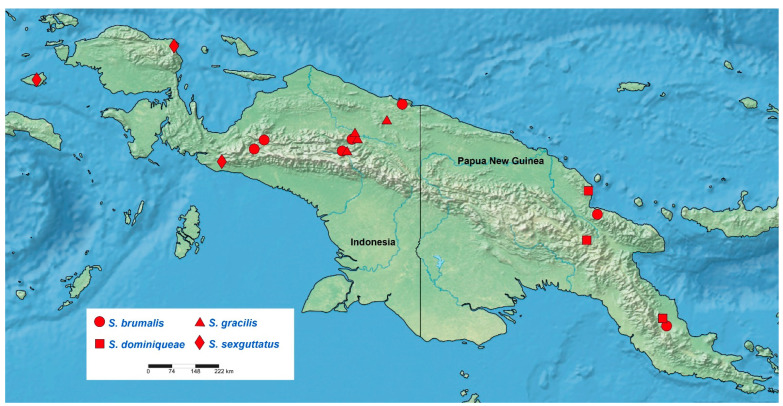
Known distribution of *Sigicoris* **stat. nov.**

## Figures and Tables

**Figure 1 insects-13-00951-f001:**
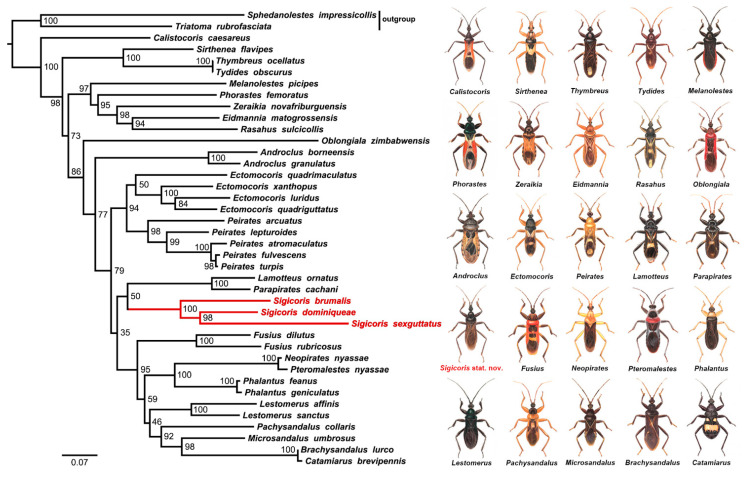
Phylogenetic tree of Peiratinae inferred via IQ-TREE based on concatenated genes (*COI* + 16S + 18S). The nodal support indicates the bootstrap support values. Red branches highlight *Sigicoris* species and beside the tree are illustrations of the peiratine genera included in phylogenetic analysis.

## Data Availability

Data are contained within the article or Appendix A.

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
