# Peer review of "Revision of the Assassin Bug Genus Sigicoris stat. nov. Based on Morphological Study and Molecular Phylogeny (Heteroptera: Reduviidae: Peiratinae)â€"

_insects, 2022, doi:10.3390/insects13100951_

Round 1
Reviewer 1 Report
This is a useful paper embodying a lot of new information, therefore worthy of publication in your journal.
However I have several major corrections/ comments / suggestions marked directly on the attached version of the manuscript, that need to be considered and addressed before the paper is accepted for publication.
See my detailed comments on the attached manuscript.

Author Response
Dear Reviewer:
Thank you so much for your valuable comments. Please see the attachment for our response.
Best wishes.

Reviewer 2 Report
The paper is very important as the main topic in a representative of Peiratinae - a subfamily of Reduviidae, which is still not well studied. Topic is original and interesting (the genus was not studied before). The paper is very well written in a standard way for this type of study. Conclusions are consistent and they resolutely address the main topic of the paper. I don’t have any comments, as the paper is very well written for this kind of study.
Author Response
Dear Reviewer:
Thank you so much for your affirmation and support to our study.
All the best to you.
Reviewer 3 Report
Journal: Insects
Manuscript Title: Revision of the assassin bug genus Sigicoris stat. nov. based on morphological study and molecular phylogeny (Heteroptera: Reduviidae: Peiratinae)
Recommendation: Minor revision needed
In this manuscript, following molecular phylogenetic and morphological analysis, Sigicoris is elevated to genus level from subgeneric status (a subgenus of Ectomocoris) and revised. This study includes a broad taxonomic sampling of peiratine genera (>70% represented) and beautiful habitus and genitalic plates for the revised/described Sigicoris species. Pending minor revision with consideration of the following comments, I would recommend this manuscript for publication in Insects.
Comments and Suggestions:
1) To further improve this manuscript, I recommend that the authors include morphological annotations across at least one of the taxon plates that correspond with the characters described in the text.
2) Could more sequence data be mined and utilized from the genomic data on hand? Today, ideally, taxonomic revisions should be forestalled until phylogenetic results are conclusive enough and support the proposed changes. With more and more insect focused phylogenetic studies employing UCE, transcriptome, and whole genome sequence data, it is only a matter of time before the same is done too extensively across Peiratinae. As only three gene regions were analyzed for this study (two mitochondrial COI + 16S and one nuclear ribosomal 18S), the support along the backbone remains quite low. Including additional loci could potentially resolve this issue. As it seems that whole genomes were sequenced de novo for many of the taxa studied, could not additional loci such as heteropteran UCEs (known from other studies with assassin bugs) be pulled out from the assemblies to provide an extensive amount of nuclear DNA for analysis? An explanation for why this approach wasn’t taken might help. Where genomes not sequence at a high enough depth to do so? Limited information on this end is provided in the Material & Methods and Results sections of the manuscript. Sigicoris appears to constitute a clade but its relation to other peiratines remains unresolved given the low support along the backbone. Here, it is recovered in a separate part of the tree than the other Ectomocoris taxa sampled but this arrangement could potentially change given the analysis of more data.
3) The authors need to clarify in the generic diagnosis section whether these are each unique characteristics of Sigicoris (which most of them are probably not) or at the beginning of the diagnosis state that “members of the genus can be recognized among Peiratinae by the following combination of characteristics:…”. In addition, it would help if there was a statement here also explicitly noting the specific differences between Sigicoris and Ectomocoris as well as members of the proposed sister group (Lamotteus + Parapirates) and any other genera for which Sigicoris might be easily confused. Some of this is given in discussion section 4.1 toward the end of the manuscript, but I think it is important to also included it in the diagnosis as many readers would first look here to verify identifications (or at least point readers to discussion section here for these important notes on distinguishing Sigicoris from other taxa).
4) For consistency, please use either “fossula spongiosa” or “spongy furrow” throughout the manuscript. Application of just one term could help eliminate any avoid potential confusion.
5) LINE 11 and LINE 37 - Change sixth largest subfamily to “one of the largest subfamilies”. The number of subfamilies within Reduviidae is in flux as several groups are currently under revision so the exact number here could change.
6) Material and methods: Please state if/how Illumina adapters were removed.
7) LINE 151 – It should read that Ectomocoris was recovered as polyphyletic with respect to Ectomocoris (Sigicoris) and not that Ectomocoris was recovered as monophyletic.
8) LINE 183 – Please elaborate on “stripes nearly invisible” as I am not sure what stripes are being referenced. It seems that the majority of peiratines have a uniformly colored anterior pronotal lobe.
9) LINE 190 – As vein names have been applied inconsistently across reduviid literature, annotating the veins discussed herein (especially Pcu) on one of the figures could clear up potential confusion. Also, should Pcu be referred to as An1 (see Schuh and Weirauch 2020)? Please clarify.
10) Some additional minor edits need to be made to the text (this list is not comprehensive, so please carefully read through text again to make modifications):
LINE 13 – use “result of molecular phylogenetic analysis using…”
LINE 23 – use “share” not “sharing”
LINE 48 – use “but are most speciose in the tropical region”
LINE 66 – use “share” not “sharing”
LINE 123 – use “The dataset, which consisted of 3,284 bp, was concatenated with…”
LINE 160 – use “belongs” not “belong”
LINES 170–172 – remove the periods for consistency
Author Response

(The authors gave the same response as above.)

Round 2
Reviewer 1 Report
Minor corrections as suggested in attached pdf (version 2)

Author Response
Dear Reviewer:
Many thanks for revising our manuscript again carefully. Please see the attachment to download our response.
Best wishes.
